# Fucoidan Improves Growth, Digestive Tract Maturation, and Gut Microbiota in Large Yellow Croaker (*Larimichthys crocea*) Larvae

**DOI:** 10.3390/nu14214504

**Published:** 2022-10-26

**Authors:** Zhaoyang Yin, Ye Gong, Yongtao Liu, Yuliang He, Chuanwei Yao, Wenxing Huang, Kangsen Mai, Qinghui Ai

**Affiliations:** 1The Key Laboratory of Aquaculture Nutrition and Feed (Ministry of Agriculture and Rural Affairs), The Key Laboratory of Mariculture (Ministry of Education), Ocean University of China, 5 Yushan Road, Qingdao 266003, China; 2Laboratory for Marine Fisheries Science and Food Production Processes, Qingdao National Laboratory for Marine Science and Technology, 1 Wenhai Road, Qingdao 266237, China

**Keywords:** early life intervention, fucoidan, large yellow croaker larvae, digestive tract maturation, gut microbiota

## Abstract

The early life period is considered an essential period for gut microbial colonization. Manipulating gut microbiota interventions during early life periods has been proven to be a promising method to boost healthy growth. Therefore, the aim of the present study was to investigate the effects of dietary fucoidan (Fuc) on the growth, digestive tract maturation, and gut microbiota of large yellow croaker (*Larimichthys crocea*) larvae. Four diets were formulated with different levels of Fuc (0.00%, 0.50%, 1.00%, and 2.00%). Results showed that dietary Fuc significantly improved the growth performance of larvae. Meanwhile, dietary Fuc promoted digestive tract maturation. Dietary 1.00% Fuc significantly improved intestinal morphology. Dietary Fuc upregulated the expression of intestinal cell proliferation and differentiation related-genes and intestinal barrier related-genes. Dietary 2.00% Fuc significantly increased the activities of brush border membranes enzymes and lipase while inhibiting α-amylase. Furthermore, dietary Fuc maintained healthy intestinal micro-ecology. In detail, dietary 1.00% and 2.00% Fuc altered the overall structure of the gut microbiota and increased the relative abundance of Bacteroidetes while decreasing the relative abundance of opportunistic pathogens and facultative anaerobe. In conclusion, appropriate dietary Fuc (1.00–2.00%) could improve the growth of large yellow croaker larvae by promoting digestive tract maturation and maintaining an ideal intestinal micro-ecology.

## 1. Introduction

The gut of fish harbors a diverse microbial community [1]. The microbial community residing in the fish gut plays a key role in the development of the host, for example: by stimulating the maturation of the host immune system [2], promoting the development of the digestive tract [3], and stimulating cell renewal in the intestinal epithelium [4]. In particular, there has been increasing evidence suggesting that the rate and trajectory of gut microbiota acquisition have a considerable impact on the health outcome of the host [5,6].

After hatching, fish larvae start accumulating gut microbiota until they are in a relatively stable state [7]. The microorganisms colonizing the gut change with the development of the fish and constantly adapt to the nutritional and environmental situation [1]. Due to an undeveloped intestinal tract and immature immune system, gut microbiota are susceptible to environmental factors during the larval phase [8]. Recently, previous studies demonstrated that nutrition intervention in early life could be a promising method for the establishment of an ideal gut microbiota [9,10,11]. However, little attention has been paid to the benefits of microorganism manipulation through dietary supplements during the period of fish larvae [12].

Marine polysaccharides have been extensively studied as a result of their beneficial functions [13]. Among marine polysaccharides, fucoidan (Fuc) has received particular attention. Fuc is a highly sulfated heterogeneous polysaccharide derived from marine brown algae and has diverse biological activities, such as anti-inflammatory [14], antioxidant [15], anti-tumor [16], anti-bacterial [17], and immunomodulatory effects [18]. Previous studies have demonstrated that dietary Fuc has a prebiotic-like effect, which could modulate the microbial community in the gastrointestinal (GI) tract [19,20]. However, studies on the effects of Fuc maintaining the intestinal health status of fish remain scarce.

The large yellow croaker (*Larimichthys crocea*) is one of the most important commercial marine fish that has been widely cultured in south China [21,22]. Similar to most marine fish species, large yellow croaker larvae are vulnerable during the metamorphosis and weaning period [12,23]. This period could be considered a “window of opportunity” for microbial colonization [10]. Therefore, our study aimed to determine the effects of dietary Fuc on the growth, digestive tract maturation, and gut microbiota of large yellow croaker larvae. Using a marine fish model, this study might provide novel insights into manipulating gut microbiota interventions during the early life period to boost the healthy growth of the host.

## 2. Materials and Methods

### 2.1. Diet Formulation

Four isonitrogenous (52% crude protein) and isolipidic (20% crude lipid) diets were formulated and supplemented with graded levels of Fuc, 0.00% (Fuc0), 0.50% (Fuc0.5), 1.00% (Fuc1), and 2.00% (Fuc2) (Appendix A). The additive Fuc form *Laminaria japonica* was purchased from Bright Moon Seaweed Group Co., Ltd. (Qingdao, China), and the effective constituent of Fuc was around 90%. Diets were manufactured by micro-bonding technology [24]. The particle size of diets ranged within 150–250 μm for fish larvae between 13 and 24 days after hatch (DAH) and 250–450 μm for fish larvae thereafter.

### 2.2. Experimental Procedure

In this present study, all experiments were performed in strict accordance with the Management Rule of Laboratory Animals (Chinese Order No. 676 of the State Council, revised 1 March 2017). The experimental procedure and design are shown in the experimental flow graph provided using Figdraw software (Figure 1).

Experiment animals used in this study were obtained from Ningde Fufa Fishery Co., Ltd. (Ningde, China). The feeding experiment was conducted at the State Key Laboratory of Large Yellow Croaker (Ningde, China). All larvae in the hatchery were fed with *Brachionus plicatilis* (0.5–1.5 × 10^4^ individual L^−1^) from 3 to 8 DAH, *Artemia nauplii* (1.0–1.5 × 10^3^ individual L^−1^) from 6 to 12 DAH, and live copepods (*Calanus sinicus*) and the experimental diets of Fuc0 group from 13 to 17 DAH. The proportion of experimental diets was gradually increased by 20% every day. Until 17 DAH, larvae were completely weaned to the experimental diets. The experiment was carried out in 12 blue plastic tanks (water volume 1000 L) at a stocking density of 10,000 larvae (17 DAH) per tank. Four experimental diets were randomly allocated to triplicate groups of larvae. The rearing temperature conditions ranged from 23 to 24 °C, pH from 7.8 to 8.2, salinity from 21 to 24‰, and photoperiod (16 h light: 8 h dark). The daily water change rate was 80–160%, and the surface water was skimmed regularly to remove the suspended waste. From 17 to 47 DAH, larvae were manually fed to satiation with the experimental diets seven times (06:00, 08:00, 11:00, 14:00, 17:00, 20:00, and 22:00) daily.

### 2.3. Sampling and Dissection

Before the experiment started, the initial body weight and length of 50 randomly collected larvae (17 DAH) from each tank were measured. At the end of the experiment, larvae were deprived of food for 24 h before sampling to empty the guts. The fish remaining in each tank at the end of the experiment were counted to determine the survival rate. One thousand larvae from each tank were randomly collected to measure final body weight and final body length. Intestinal segments (IS) and pancreatic segments (PS) of 80 larvae were separated under a dissecting microscope at 0 °C for digestive enzyme activities assays [25]. Fifty larvae from each tank were dissected under a dissecting microscope to obtain the intestine and immediately frozen in liquid nitrogen for intestinal gene expression assays. Thirty larvae were sampled from each tank and preserved in 4% paraformaldehyde for 24 h and transferred to 75% alcohol for the analysis of intestinal morphology. The whole intestines of 50 larvae from each tank were separated aseptically under a dissecting microscope for the analysis of gut microbiota.

### 2.4. Analytical Methods

#### 2.4.1. Intestinal Histology Analysis

The intestinal micromorphology was determined based on the method described by previous research [26]. Briefly, the larvae were washed and dehydrated with gradient alcohol, and then larvae were paraffin-embedded, sectioned, and stained with hematoxylin and eosin. All images were analyzed using Image-Pro Plus 6.0 software (Version 6.0, Media Cybernetics, Rockville, MD, USA) to measure villus height, villus width, and muscular thickness (the maximum length of each villus from top to the root was defined as villus height, the maximum width of each villus was defined as villus width, and the distance between the epicuticle and endocuticle was defined as muscular thickness).

#### 2.4.2. cDNA Synthesis and Real-Time Quantitative Polymerase Chain Reaction (qPCR)

Total RNA was extracted from the visceral mass or intestine using RNAiso Plus (Takara, Kyoto, Japan). The quality and integrity of RNA were measured by electrophoresis using 1.2% denatured agarose gel and then assessed by the Nano Drop^®^2000 spectrophotometer (Thermo Fisher Scientific, Waltham, MA, USA) to test the concentration. Then, RNA were reverse transcribed to cDNA using the Prime Script-RT reagent Kit (Takara, Japan). The real-time quantitative polymerase chain reaction was carried out in a quantitative thermal cycler (CFX96TM Real-Time System, Bio-Ras, Hercules, CA, USA). The real-time quantitative PCR temperature profile was 95 °C for 2 min, followed by 35 cycles of 95 °C for 10 s, 58 °C for 10 s, and 72 °C for 20 s. At the end of each PCR reaction, melting curve analysis was performed to confirm that a single PCR product was present in each one of these reactions. The primer sequences were designed and synthesized based on the published sequences from GenBank (Appendix A) [23,27]. The fluorescence data acquired during the extension phase were normalized to β-actin via 2^−ΔΔCT^ methods.

#### 2.4.3. Digestive Enzyme Activities Assay

PS and IS (0.2 g) of 22, 27, 37, and 47 DAH larvae were weighed and homogenized in 1.8 mL 0 °C normal saline, then centrifuged at 3000× *g* for 10 min, and the supernatant was collected for further assay. Purified brush border membranes (BBM) from the homogenate of IS were obtained according to a published paper [28]. The activity of leucine-aminopeptidase (LAP) used leucine-*p*-nitroanolide as a substrate [29]. The activity of α-amylase, lipase, trypsin, and alkaline phosphatase (AKP) was examined using commercial assay kits (Nanjing Jiancheng Bioengineering Institute, Nanjing, China).

#### 2.4.4. Microbiota Analyses

Microbial DNA was extracted from the whole intestine of 47 DAH larvae using the CTAB method. Briefly, the intestinal tissue of 50 larvae was dissolved in a CTAB lysis solution, then phenol:chloroform:isoamyl alcohol (25:24:1) solution was added and centrifuged at 12,000× *g* for 10 min. The supernatant was collected, then chloroformisoamyl alcohol (24:1) was added and centrifuged at 12,000× *g* for 10 min. Isoamyl alcohol was added into the collected supernatant, then centrifuged at 12,000× *g* for 10 min. Collected DNA sediment was washed with 75% ethyl alcohol and then dissolved DNA sediment using ddH_2_O. RNase A solution (1 μL) was used for slaking RNA. Successful DNA extraction was confirmed by 1.0% agarose gel electrophoresis. The V4-V5 hypervariable region of the bacterial 16S rRNA gene was amplified using barcoded primers: Fwd5′-GTGCCAGCMGCCGCGGTAA-3′ and Rev5′-CCGTCAATTCCTTTGAGTTT-3′. The PCR conditions were pre-denaturation at 98 °C for 1 min, 30 cycles of denaturation at 98 °C for 10 s, annealing at 50 °C for 30 s, elongation at 72 °C for 30 s, and a final post-elongation cycle at 72 °C for 5 min. Then, PCR products were purified with a Qiagen Gel Extraction Kit (Qiagen, Dusseldorf, Germany). After purification, the PCR products were used for the construction of libraries and then paired-end sequenced on an Illumina MiSeq platform provided by Beijing Novogene Genomics Technology Co., Ltd. (Beijing, China). The complete data was submitted to the NCBI Sequence Read Archive (SRA) database under accession number PRJNA735925.

After being assembled, quality screened, and trimmed, a total of 86,048 high-quality valid reads was obtained, resulting in the identification of 1781 operational taxonomic units (OTUs) with 97% identity from all samples [30]. For all samples, the rarefied curves for the observed species number tended to approach the saturation plateau, indicating complete sequencing efforts for all samples (Appendix A). The α-diversity indices (observed species, Chao1, ACE, PD whole tree, Shannon, and Simpson) of bacterial richness and diversity were calculated with QIIME (Version 1.7.0). The β-diversity was analyzed by principal component analysis (PCA) based on OTUs, and the hierarchical clustering tree was constructed based on unweighted Unifrac distances. To assess the changes in microbial community structure, differentially abundant taxa among treatments were identified by the linear discriminant analysis effect size (LEfSe) analysis [31]. The microbial phenotypes among treatments were predicted with the BugBase software package [32]. Spearman’s correlation between gut microbiota and the selected intestinal gene markers was determined using R packages (Version 2.15.3).

### 2.5. Calculations and Statistical Analysis

Before analysis, data from each treatment were tested for normality and variance homogeneity using the Kolmogorov–Smirnov test and the Bartlett test, respectively. Statistical analysis was performed in SPSS 16.0 (SPSS Inc., Chicago, IL, USA). Data from each treatment were subjected to one-way analysis of variance (ANOVA) followed by Tukey’s test or the pairwise Mann–Whitney–Wilcoxon test. For statistically significant differences, *p* < 0.05 was applied. Results were presented as mean ± S.E.M. (standard error of means).

The growth parameters were calculated as follows:Survival rate (%) = N_t_ × 100/N_i_,(1)
Weight gain rate (%) = (W_t_ − W_i_)/W_i_,(2)
Specific growth rate (% day^−1^) = (LnW_t_ − LnW_i_) × 100/d,(3)
where N_t_ is the final number of larvae in each tank and N_i_ is the initial number of larvae in each tank at the beginning of the experiment; W_t_ and W_i_ are the final and initial body weight, respectively; d is the experimental period in days.

## 3. Results

### 3.1. Dietary Fuc Improved the Growth of Larvae

Compared with the Fuc0 group, dietary Fuc significantly improved the final body weight, final body length, weight gain rate, and specific growth rate of large yellow croaker larvae (*p* < 0.05) (Table 1). Furthermore, no significant difference in survival rate was observed among dietary treatments (*p* > 0.05) (Table 1).

### 3.2. Dietary Fuc Promoted Maturation of the Digestive Tract

#### 3.2.1. Dietary Fuc Improved Intestinal Morphology

Dietary Fuc effectively improved the intestinal morphology of large yellow croaker larvae (Appendix A and Table 2). Larvae fed diets with Fuc had significantly higher villus height than the Fuc0 group (*p* < 0.05), and the maximum villus height of the intestinal tract was observed in the Fuc1 group (Table 2). Similarly, significantly higher muscular thickness and villus width were observed in larvae-fed diets with 1.00% Fuc compared to the Fuc0 group (*p* < 0.05) (Table 2).

#### 3.2.2. Dietary Fuc Regulated Intestinal Cell Proliferation, Differentiation-Related Genes, and Intestinal Barrier-Related Gene Expression

Larvae fed diets with 0.50% Fuc had significantly higher mRNA expression of *pcna* and *odc* compared to the Fuc0 group (*p* < 0.05) (Figure 2a). With increasing dietary Fuc, the *akp* transcriptional level increased first and then decreased, and the maximum gene expression of *akp* was observed in the Fuc0.5 group (Figure 2a). Similarly, larvae fed diets with Fuc had significantly higher mRNA expression of *zo-1* and *claudin-11* compared to the Fuc0 group (*p* < 0.05) (Figure 2b). Larvae fed the diet with 0.50% Fuc had a significantly higher *zo-2* transcriptional level compared to the Fuc0 group (*p* < 0.05) (Figure 2b). Meanwhile, significantly higher *occludin* mRNA expression was observed in larvae fed the diet with 2.00% Fuc compared to the Fuc0 group (*p* < 0.05) (Figure 2b).

#### 3.2.3. Long-Term Dietary Fuc Improved Larval Digestive Functions

Four stages (22, 27, 37, and 47 DAH) of large yellow croaker larvae were chosen to assess the short-term and long-term effects of dietary Fuc on larval digestive functions (Table 3 and Table 4). Results showed no significant difference in the activity of AKP, LAP, trypsin, lipase, and α-amylase of larvae at both 22 DAH and 27 DAH among dietary treatments (*p* > 0.05), indicating that short-term dietary Fuc supplementation could not improve the larval digestive functions of the large yellow croaker (Table 3 and Table 4).

In the 37 DAH phase, the activity of AKP was gradually increased among dietary treatments and was significantly higher in the Fuc groups than in the Fuc0 group (*p* < 0.05) (Table 3). Meanwhile, larvae fed the diet with 2.00% Fuc significantly increased the activity of LAP compared to the Fuc0 group (*p* < 0.05) (Table 3). The lipase activity of larvae was gradually increased and thereafter decreased with increasing Fuc in the diet (Table 4). Larvae fed the diet with 1.00% Fuc significantly increased the activity of lipase in PS and IS compared to the Fuc0 group (*p* < 0.05) (Table 4). Dietary Fuc reduced the activity of α-amylase in PS and IS, and the maximum activity of α-amylase was observed in the Fuc0 group (Table 4).

In the 47 DAH phase, the activities of AKP and LAP in BBM were gradually increased with increasing dietary Fuc, and the highest activities of AKP and LAP were observed in the Fuc2 group (*p* < 0.05) (Table 3). Dietary Fuc significantly increased the activity of lipase in IS compared to the Fuc0 group (*p* < 0.05) (Table 4). Meanwhile, larvae fed with 0.50% and 1.00% Fuc increased the activity of lipase in PS more than in the Fuc0 group (*p* < 0.05) (Table 4). Moreover, dietary Fuc decreased the activity of α-amylase (Table 4).

### 3.3. Dietary Fuc Improved Larval Gut Microbiota

#### 3.3.1. Dietary Fuc Altered the Overall Structure of the Gut Microbiota

Results showed that the number of unique OTUs in the Fuc0, Fuc0.5, Fuc1, and Fuc2 groups was 41, 23, 96, and 30, respectively (Figure 3a). The α-diversity indexs results indicated that diversity estimates and richness estimates of larval gut microbiota decreased firstly and then increased with increasing dietary Fuc, but there were no significant differences between the Fuc0 group and other Fuc groups (*p* > 0.05) (Appendix A). To analyze the extent of similarities in microbial communities, PCA was conducted to determine β-diversity (Figure 3b). The PCA results showed that Fuc1 and Fuc2 groups were all separated from the Fuc0 group, whereas the Fuc0.5 group had more similarity with the Fuc0 group (Figure 3b). Similarly, the results of the hierarchical clustering tree showed that the gut microbiota structure from the Fuc1 group and the Fuc2 group were similar and clustered within one higher branch, whereas the Fuc0 group and Fuc0.5 group were similar and formed another branch (Figure 3c).

#### 3.3.2. Dietary Fuc Modulated Gut Microbiota Composition

At the phylum level, the dominant bacterial communities were Proteobacteria, Firmicutes, and Bacteroidetes in the intestinal tract of larvae from all groups (Figure 4a). At the genus level, the dominant bacteria were classified into *Leisingera*, *Alcaligenes*, and *Acinetobacter* (Figure 4b). At the species level, *Alcaligenes faecalis*, *Acinetobacter lwoffii*, and *Pseudomonas psychrotolerans* composed the dominant species of larval gut microbiota communities (Figure 3c). To further study the gut microbial community composition in response to Fuc treatment, LEfSe analysis was performed (Figure 4d,e). Compared with the Fuc0 group, dietary Fuc significantly downregulated the relative abundance of order Enterobacteriales, family Enterobacteriaceae, genus *Serratia*, and species *Serratia marcescens* and *Pseudomonas psychrotolerans* (*p* < 0.05) (Figure 4d). Dietary Fuc at 0.50% significantly increased the relative abundance of the family Pseudomonadaceae and genus *Pseudomonas* (*p* < 0.05) (Figure 4d). Dietary Fuc at 1.00% significantly increased the relative abundance of phylum Bacteroidetes and class Bacteroidia (*p* < 0.05) (Figure 4d). Meanwhile, dietary Fuc at 2.00% significantly increased the relative abundance of class Bacilli, order Flavobacteriales families unidentified Flavobacteriaceae and Moraxellaceae, and genus *Acinetobacter* (*p* < 0.05) (Figure 4d).

#### 3.3.3. Dietary Fuc Affected Oxygen Utilization of the Microbial Community

In order to further understand the structure of gut microbiota in response to Fuc supplementation, BugBase analysis was performed to predict microbiome phenotypes (Figure 5). Compared with the Fuc0 group, the inclusion of 1.00% and 2.00% Fuc in diets decreased the relative abundance of facultatively anaerobic bacteria (*p* < 0.05) (Figure 5a). Meanwhile, the relative abundance of anaerobic bacteria decreased first and then increased with increasing dietary Fuc, and the minimum relative abundance of anaerobe was observed in the Fuc0.5 group (Figure 5b). However, no significant differences in the relative abundance of aerobic bacteria were observed among dietary treatments (*p* > 0.05) (Figure 5c).

#### 3.3.4. The Association between Gut Microbiota and the Selected Intestinal Gene Markers

To determine the potential association between the gut microbiota and the selected intestinal gene markers, Spearman’s correlation analysis was performed (Figure 6). The results showed that *Acinetobacter lwoffii*, *Pseudomonas psychrotolerans*, *Serratia marcescens*, *Achromobacter xylosoxidans*, and *Moraxella osloensis* were closely associated with the selected gene markers (Figure 6). To be specific, the decreasing trend of *Achromobacter xylosoxidans*, *Serratia marcescens*, and *Pseudomonas psychrotolerans* was strongly positively correlated with the expression of the selected gene markers for the intestinal barrier (Figure 6). Meanwhile, the decreasing trend of *Moraxella osloensis* was strongly positively correlated with the gene expression of the selected markers for epithelial proliferation and differentiation (*pcna* and *odc*) (Figure 6). The increase in *Acinetobacter lwoffii* in Fuc treatments was strongly positively correlated with the *occludin* transcriptional level (Figure 6).

## 4. Discussion

### 4.1. Dietary Fuc Boosted Healthy Growth of Fish Larvae

Previous studies have suggested the gut microbiota as a potential solution to boost healthy neonatal growth [6]. The establishment of an ideal gut microbiota could improve immature physical characteristics during the early life period, such as stimulating the development of the immune system [33], accelerating the maturation of the GI tract [34], and regulating the metabolism of the host [3]. In the present study, dietary Fuc was found to have positive effects on the improvement of growth performance in large yellow croaker larvae. However, the mechanisms that underpin the growth-promoting effect of Fuc were only partially elucidated. Combined with the results of the present study and current research, we speculated that dietary Fuc could partly improve growth in large yellow croaker larvae by accelerating digestive tract maturation and regulating the gut microbiota profile [35,36,37]. In addition, Fuc was proven to be a potential activator of NRF2 (nuclear factor-E2 related factor 2) and had prominent antioxidant activity [38]. The current study also found that dietary Fuc ameliorated the oxidative stress of the large yellow croaker caused by extremely rapid growth during the larval period (Appendix A).

### 4.2. Dietary Fuc Promoted Maturation of the Digestive Tract

Marine fish larvae undergo major cellular, morphological, and functional changes in the digestive tract during the early life period [39]. The maturational process of the digestive tract can be influenced depending on diet composition [40]. Thus, the present study further investigated the effects of dietary Fuc on the maturation of the digestive tract. In this work, dietary 1.00% Fuc significantly increased villus height, villus width, and muscular thickness. Notably, the maturation of the larval intestine and the improvement of intestinal morphology were closely related to the proliferation and differentiation of intestinal epithelial cells [41,42]. In the present study, the selected markers for epithelial proliferation and differentiation were highly expressed in Fuc treatments. Similarly, dietary Fuc upregulated the gene expression of intestinal barrier, which indicated dietary Fuc could enhance intestinal barrier function in large yellow croaker larvae. Interestingly, the GI microbiome is involved in epithelial differentiation and maturation [12]. For example, the GI microbiome of zebrafish larvae was found to stimulate epithelial proliferation by increasing the expression of 15 genes involved in DNA replication and cell division [34]. In the present study, the relative abundance of opportunistic pathogens (*Achromobacter xylosoxidans*, *Serratia marcescens*, *Pseudomonas psychrotolerans*, and *Moraxella osloensis*) was reduced in Fuc treatments which were closely associated with the high expression of the selected gene markers. These results indicated that dietary Fuc could decrease opportunistic pathogens, resulting in maintaining intestinal barrier function and promoting epithelial differentiation and maturation.

Digestive enzymes were widely used as markers of the development rate, food acceptance, and digestive capacity of fish larvae [43]. As a promising candidate for marine drugs, the effects of Fuc on the metabolism have been well illustrated [44]. Many potential mechanisms, including the activation of lipase and inhibition of α-amylase and α-glucosidase, have been proposed [20,45]. Similarly, the present study found that dietary Fuc increased the activity of lipase but decreased the activity of α-amylase in 37 DAH and 47 DAH larvae. Furthermore, the establishment of an efficient BBM digestion mechanism represents the adult digestive model [46]. The onset of intestinal BBM enzymes (AKP and LAP) has been widely used to determine the maturation process of the digestive system in fish larvae. In this work, dietary 2.00% Fuc improved the activities of LAP and AKP in the BBM of 37 DAH and 47 DAH larvae, which demonstrated that dietary Fuc could promote the switch of digestion from a primary to an adult mode. Interestingly, there has been increasing evidence suggesting that the absence of the gut microbiome hampers epithelial differentiation and function of the GI tract, such as the lack of AKP activity in BBM [47]. Therefore, dietary Fuc could increase the activities of intestinal BBM enzymes by modulating the gut microbiota of fish larvae. However, the current study indicated that short-term dietary Fuc could not improve larval digestive functions. The results suggested that early life intervention should consider the timing, dosage, and duration, which might consequently lead to divergent outcomes.

### 4.3. Dietary Fuc Improved the Gut Microbiota of Fish Larvae

The intestines of fish can harbor 10^7^ to 10^11^ bacteria/g intestinal content, and the complex intestinal microbial communities are implicated in a great number of host functions [48,49]. However, the gut microbiota is not fully assembled in the larval gut tract during the early life period [4]. A previous study validated the early life period and considered it an ideal window for gut microbial colonization [50,51]. Therefore, this study, for the first time, focused on the effects of dietary Fuc on the gut microbiota of large yellow croaker larvae. The previous study suggested that there was a structure-dependent relationship between dietary Fuc and modulation of the gut microbiota [20]. In the present study, results showed that dietary Fuc had a strong effect on the overall structure of the intestinal microbiota of large yellow croaker larvae. Interestingly, the effect of Fuc on the gut microbiota structure was dose-dependent. The results of PCA and hierarchical clustering tree analysis showed that the Fuc1 and Fuc2 groups were separated from the Fuc0 group, whereas the Fuc0.5 group had more similarity with the Fuc0 group. The above results indicated that 1.00% and 2.00% dietary Fuc might have a more profound effect on the intestinal microbial community in larvae than 0.50% Fuc.

Recent studies showed that the microbiome of fish larvae consisted of four dominant phyla: Proteobacteria, Bacteroidetes, Firmicutes, and Actinomycetes, with changes in dominance usually observed depending on the nutritional and environmental situation [12]. In line with these studies, the most abundant phyla detected in the present study were Proteobacteria, Firmicutes, and Bacteroidetes in all groups. The current data showed that the relative abundance of the core phylum Bacteroidetes increased in the Fuc1 and Fuc2 groups. Bacteroidetes is the major group of complex polysaccharide degraders, which can utilize non-digestible carbohydrates and produce beneficial bacterial metabolites [44,52]. In accordance with previous studies, Fuc could possibly be fermented by Bacteroidetes. BugBase analysis revealed a decrease in facultative anaerobes after Fuc treatments, indicating that dietary Fuc altered oxygen utilization of the microbial community, which might contribute to the colonization of commensal microflora [33]. The previous studies indicated that facultative anaerobes contain many pathogenic bacteria, including *Escherichia coli*, *Serratia*, *Salmonella*, *Edwardsiella tarda*, and *Aeromonas hydrophila* [20]. In addition, the results of the lefSe analyses showed significant decreases in pathogens in Fuc treatments. Potential pathogens, such as Enterobacteriaceae, *Serratia marcescens*, and *Pseudomonas psychrotolerans*, were significantly decreased in Fuc treatments, indicating that dietary Fuc may prevent pathogenic bacteria from invading the intestinal mucosa [44]. The above results suggested that early life Fuc intervention maintained healthy intestinal microecology, including increasing the relative abundance of polysaccharides degraders while decreasing that of opportunistic pathogens and facultative anaerobes, which might contribute to boosting the healthy growth of fish larvae.

## 5. Conclusions

In conclusion, the present study found that early life dietary Fuc intervention had a potentially positive effect on the growth, digestive tract maturation, and gut microbiota profile of large yellow croaker larvae. These effects were dose-dependent and time-dependent. Under the present experimental conditions, Fuc is a promising prebiotics candidate and its optimum dosage is 1.00–2.00%.

## Figures and Tables

**Figure 1 nutrients-14-04504-f001:**
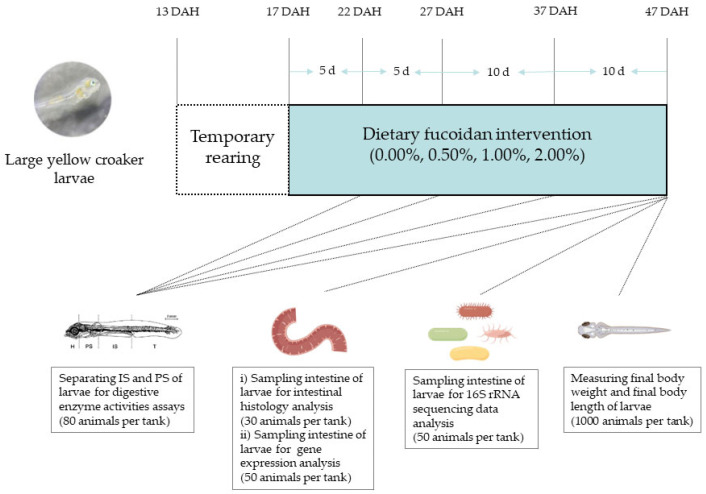
Flow graph of experimental procedure and design (DAH: days after hatch; IS: intestinal segments; PS: pancreatic segments).

**Figure 2 nutrients-14-04504-f002:**
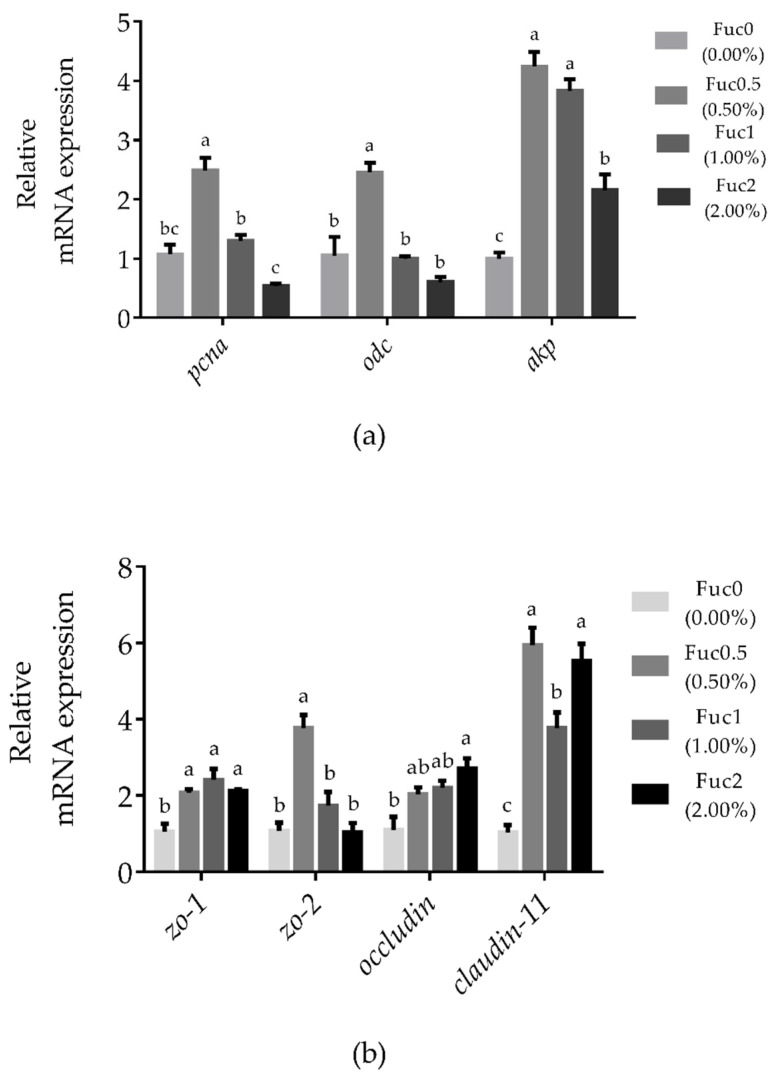
Effects of dietary fucoidan on intestinal cell proliferation and differentiation-related genes mRNA expression (**a**) and intestinal barrier-related genes mRNA expression (**b**) in the intestinal tract of large yellow croaker larvae. Values are means (n = 3), with their standard errors represented by vertical bars. Bars bearing the same letters were not significantly different (*p* > 0.05, Tukey’s test).

**Figure 3 nutrients-14-04504-f003:**
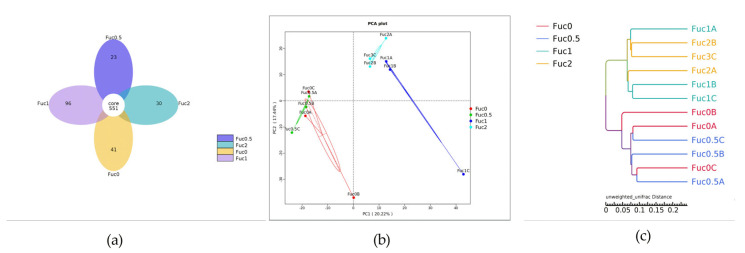
Effects of dietary fucoidan on gut microbial structure of large yellow croaker larvae (n = 3/group). (**a**) Venn diagram; (**b**) principal component analysis (PCA); (**c**) unweighted uniFrac distance matrix.

**Figure 4 nutrients-14-04504-f004:**
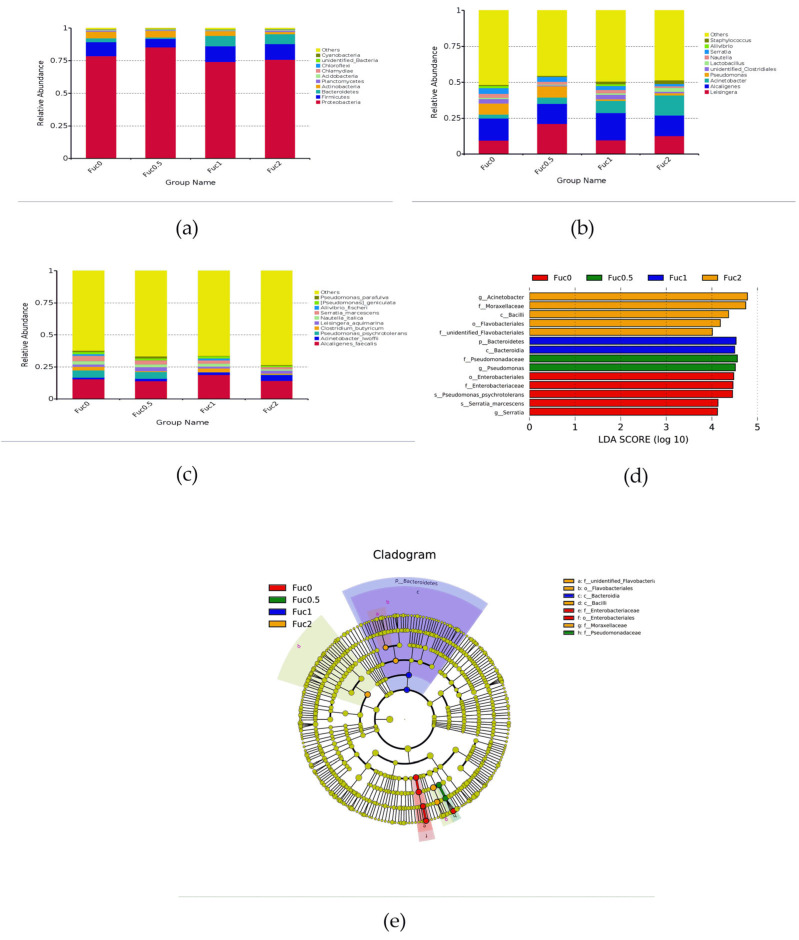
Effects of dietary fucoidan on gut microbial composition of large yellow croaker larvae (n = 3/group). (**a**–**c**) Taxonomy classification of reads at phylum (**a**), genus (**b**) and specie (**c**) taxonomic levels. Only the top 10 most abundant (Based on relative abundance) bacterial phyla, genera and species were shown in the figures. Other phyla, genera and species were all assigned as ‘Others’. (**d**,**e**) LefSe analysis identified the most differentially abundant taxons among the Fuc0, Fuc0.5, Fuc1, and Fuc2 groups.

**Figure 5 nutrients-14-04504-f005:**
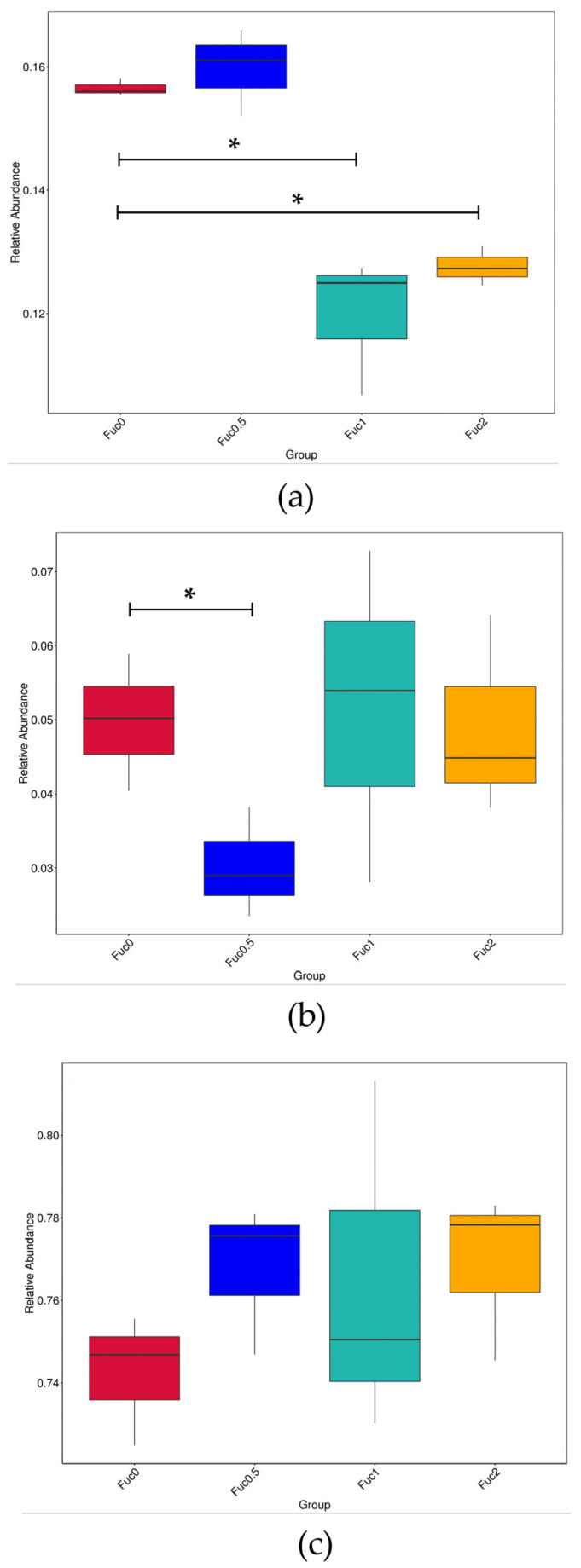
Effects of dietary fucoidan on oxygen utilization of microbial community of large yellow croaker larvae based on BugBase analysis. (**a**) Facultatively anaerobic bacteria; (**b**) anaerobic bacteria; (**c**) aerobic bacteria. The outcome was grouped according the modules (*x*-axis). The relative abundance is presented on the *y*-axis. Pairwise Mann–Whitney–Wilcoxon tests were performed for data analysis between the Fuc0 group and other Fuc groups (*, *p* < 0.05).

**Figure 6 nutrients-14-04504-f006:**
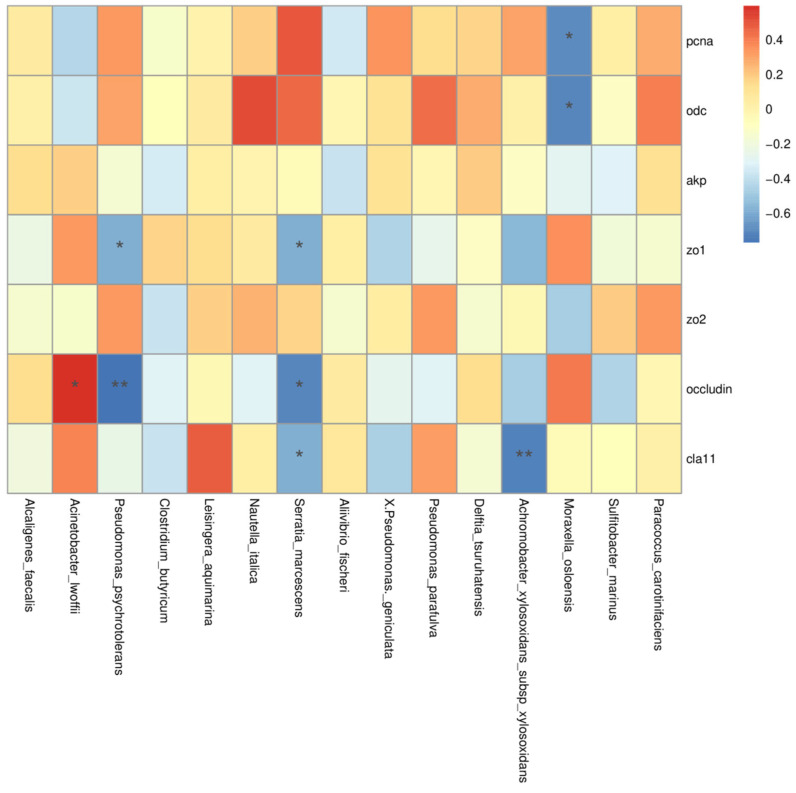
Effects of dietary fucoidan on the relative abundance of gut microbiota that correlated with the selected intestinal gene markers (zo1: tight zonula occludens-1; zo2: tight zonula occludens-2; occludin; cla11: claudin-11; pcna: proliferating cell nuclear antigen; odc: ornithine decarboxylase; akp: alkaline phosphatase). Pearson’s correlation with the false discovery rate (FDR) of the relative abundance of gut microbiota and the selected intestinal gene markers (*, *p* < 0.05; **, *p* < 0.01).

**Table 1 nutrients-14-04504-t001:** Effects of dietary fucoidan on survival and growth of large yellow croaker larvae (Means ± S.E.M., n = 3) ^1^.

Parameters	Experimental Diets (Fuc%)			
	Fuc0 (0.00%)	Fuc0.5 (0.50%)	Fuc1 (1.00%)	Fuc2 (2.00%)
Initial weight (mg)	3.45 ± 0.20	3.37 ± 0.13	3.38 ± 0.09	3.14 ± 0.09
Final weight (mg)	81.52 ± 2.97 ^b^	102.77 ± 7.09 ^a^	101.53 ± 2.97 ^a^	106.14 ± 2.39 ^a^
Initial length (mm)	6.00 ± 0.07	5.94 ± 0.07	5.83 ± 0.04	5.87 ± 0.15
Final length (mm)	16.20 ± 0.13 ^b^	18.02 ± 0.10 ^a^	17.90 ± 0.32 ^a^	17.51 ± 0.20 ^a^
Survival rate (%)	23.30 ± 4.91	18.13 ± 3.85	26.75 ± 4.58	26.02 ± 2.11
Weight gain rate (%)	2270.87 ± 97.38 ^b^	2945.55 ± 147.06 ^a^	2909.42 ± 161.38 ^a^	3277.67 ± 60.01 ^a^
Specific growth rate (%/day)	10.55 ± 0.14 ^b^	11.38 ± 0.16 ^a^	11.34 ± 0.17 ^a^	11.73 ± 0.06 ^a^

^1^ Data in the same row sharing the same superscript letter are not significantly different, determined by Tukey’s test (*p* > 0.05).

**Table 2 nutrients-14-04504-t002:** Effects of dietary fucoidan on morphology of the intestine of large yellow croaker larvae (Means ± S.E.M., n = 3) ^1^.

Parameters	Experimental Diets (Fuc%)			
	Fuc0 (0.00%)	Fuc0.5 (0.50%)	Fuc1 (1.00%)	Fuc2 (2.00%)
Villus height (μm)	85.67 ± 4.17 ^c^	125.86 ± 10.16 ^b^	156.85 ± 3.04 ^a^	118.52 ± 6.91 ^b^
Villus width (μm)	40.28 ± 2.36 ^b^	57.60 ± 5.27 ^ab^	75.71 ± 4.27 ^a^	59.27 ± 9.42 ^ab^
Muscular thickness (μm)	14.57 ± 0.66 ^b^	23.67 ± 2.87 ^ab^	28.11 ± 4.04 ^a^	25.82 ± 2.46 ^ab^

^1^ Data in the same row sharing the same superscript letter are not significantly different, determined by Tukey’s test (*p* > 0.05).

**Table 3 nutrients-14-04504-t003:** Effects of dietary fucoidan on activities of alkaline-phosphatase and leucine-aminopeptidase in brush border membrane (BBM) of large yellow croaker larvae (Means ± S.E.M., n = 3) ^1^.

Parameters	DAH ^3^	Experimental Diets (Fuc%)			
		Fuc0	Fuc0.5	Fuc1	Fuc2
		0.00%	0.50%	1.00%	2.00%
AKP ^2,3^	22	1583.22 ± 269.21	1474.55 ± 179.29	1497.24 ± 221.74	1985.62 ± 238.67
	27	1517.57 ± 333.17	2916.42 ± 282.13	2784.35 ± 451.22	3774.64 ± 959.48
	37	3446.03 ± 152.81 ^c^	4294.91 ± 120.97 ^b^	4624.98 ± 82.03 ^b^	5414.87 ± 190.84 ^a^
	47	4456.80 ± 463.26 ^c^	5453.86 ± 183.89 ^bc^	6060.50 ± 207.81 ^ab^	7239.89 ± 336.34 ^a^
LAP ^2,3^	22	4.39 ± 0.58	4.12 ± 0.76	3.81 ± 0.51	4.80 ± 0.45
	27	12.98 ± 2.15	9.81 ± 1.59	10.97 ± 1.13	10.06 ± 0.90
	37	8.02 ± 1.17 ^b^	17.68 ± 2.12 ^ab^	18.93 ± 2.72 ^ab^	28.18 ± 5.67 ^a^
	47	30.39 ± 3.45 ^b^	37.98 ± 0.62 ^b^	39.28 ± 1.55 ^b^	51.56 ± 2.13 ^a^

^1^ Data in the same row sharing the same superscript letter are not significantly different, determined by Tukey’s test (*p* > 0.05). ^2^ The units of enzyme activity are mU/mg·protein. ^3^ DAH: days after hatch; AKP: alkaline-phosphatase; LAP: leucine-aminopeptidase.

**Table 4 nutrients-14-04504-t004:** Effects of dietary fucoidan on activities of digestive enzyme of large yellow croaker larvae (Means ± S.E.M., n = 3) ^1^.

Parameters	Tissue	DAH ^4^	Experimental Diets (Fuc%)			
			Fuc0	Fuc0.5	Fuc1	Fuc2
			0.00%	0.50%	1.00%	2.00%
Lipase ^2,4^	PS ^4^	22	3.80 ± 1.40	4.54 ± 1.00	4.28 ± 1.09	3.10 ± 0.16
		27	4.02 ± 0.31	4.14 ± 0.15	4.81 ± 0.18	4.31 ± 0.12
		37	3.98 ± 0.74 ^b^	5.28 ± 0.30 ^ab^	6.53 ± 0.54 ^a^	5.26 ± 0.15 ^ab^
		47	1.77 ± 0.10 ^c^	2.85 ± 0.04 ^b^	3.87 ± 0.29 ^a^	2.16 ± 0.31 ^bc^
	IS ^4^	22	2.33 ± 1.10	3.27 ± 0.87	3.14 ± 0.25	2.57 ± 0.19
		27	4.37 ± 1.04	4.11 ± 0.68	4.96 ± 0.68	3.64 ± 0.41
		37	6.02 ± 0.12 ^c^	8.16 ± 0.13 ^a^	7.85 ± 0.67 ^ab^	6.21 ± 0.27 ^bc^
		47	1.96 ± 0.24 ^b^	5.48 ± 0.09 ^a^	5.63 ± 0.50 ^a^	6.26 ± 0.20 ^a^
Trypsin ^3,4^	PS ^4^	22	1.28 ± 0.05	1.11 ± 0.04	1.13 ± 0.07	1.34 ± 0.04
		27	2.13 ± 0.15	2.50 ± 0.18	2.12 ± 0.06	2.18 ± 0.12
		37	2.21 ± 0.18	2.37 ± 0.13	2.19 ± 0.05	2.11 ± 0.10
		47	3.18 ± 0.12	2.16 ± 0.25	2.83 ± 0.48	2.91 ± 0.44
	IS ^4^	22	5.84 ± 1.12	4.48 ± 0.39	3.88 ± 0.31	5.59 ± 0.68
		27	1.52 ± 0.19	1.42 ± 0.04	1.36 ± 0.05	1.72 ± 0.14
		37	7.08 ± 0.99	6.81 ± 0.73	5.43 ± 0.99	5.73 ± 0.60
		47	10.48 ± 1.14	8.78 ± 2.13	10.80 ± 2.13	8.60 ± 0.26
α-amylase ^3,4^	PS ^4^	22	0.32 ± 0.02	0.32 ± 0.00	0.31 ± 0.02	0.31 ± 0.02
		27	0.25 ± 0.04	0.24 ± 0.01	0.24 ± 0.04	0.22 ± 0.03
		37	0.68 ± 0.07 ^a^	0.37 ± 0.05 ^b^	0.40 ± 0.02 ^b^	0.47 ± 0.02 ^b^
		47	0.55 ± 0.01 ^a^	0.32 ± 0.01 ^c^	0.48 ± 0.01 ^a^	0.43 ± 0.04 ^ab^
	IS ^4^	22	0.57 ± 0.05	0.52 ± 0.04	0.63 ± 0.04	0.54 ± 0.02
		27	0.26 ± 0.05 ^ab^	0.20 ± 0.02 ^ab^	0.12 ± 0.00 ^b^	0.27 ± 0.04 ^a^
		37	0.68 ± 0.03 ^a^	0.57 ± 0.03 ^ab^	0.44 ± 0.04 ^bc^	0.31 ± 0.04 ^c^
		47	0.57 ± 0.02 ^a^	0.45 ± 0.02 ^bc^	0.41 ± 0.02 ^c^	0.50 ± 0.01 ^ab^

^1^ Data in the same row sharing the same superscript letter are not significantly different, determined by Tukey’s test (*p* > 0.05). ^2^ The units of enzyme activity are U/g·protein. ^3^ The units of enzyme activity are U/mg·protein. ^4^ DAH: days after hatch; PS: pancreatic segments; IS: intestinal segments.

## Data Availability

Not applicable.

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
