# Peer review of "Fucoidan Improves Growth, Digestive Tract Maturation, and Gut Microbiota in Large Yellow Croaker (Larimichthys crocea) Larvae"

_nutrients, 2022, doi:10.3390/nu14214504_

Round 1
Reviewer 1 Report
Dear authors,
Please, find below major and minor suggestion for improvement of your manuscript, entitled: 'Fucoidan (Fuc) improves growth, digestive tract maturation, and gut microbiota in large yellow croaker (Larimichthys crocea) Larvae'.
Your study is showing importance of the dietary modulation of gut microbiota for development of gastrointestinal tract.
Major comments
1. The experimental flow must be presented in form of schematic illustration, showing total amount of animals, division into the groups, introducing the diets, material collection (e.g. for which method etc). Since you use 12 tanks, you have triplicated the groups of larvae for each treatment. Nevertheless, for the results and statistic you are using only 3 animals/group, leading readers to think that you analysed only 1 animal per tank/per group. This number is insufficient for the statistics, results interpretation. If you used 10 animal per groups from 3 different tanks, you should indicate the total amount of animals, e.g. n=30.
2. Trypsin analysis has been performed in pancreatic tissue and in small intestinal homogenates. Trypsin is produced in form of pro-enzyme (trypsinogen) in the exocrine pancreas, while in small intestine is already activated to its active form. Please, specify what form of trypsin have you analysed.
3. I highly recommend to divide Figure 1 into Fig 1A and Fig 1B, showing proliferation/differentiation-related genes and intestinal barrier-related genes, respectively.
4. Quality of Figure 2 and 3 must be improved.
Minor comments:
1. Do not use abbreviation in the title. Remove (Fuc), line 2.
2. Please, use official e-mail addresses as a contact information.
3. Do not use word ‘micro’ diets, it would be okej to say ‘diets’ instead.
4. You use double name for your groups, e.g. Fuc1, Fuc2, Fuc3 or Fuc 0.5%, Fuc 1.00% and Fuc 2.00%). May be introducing only one name at the experimental design and keeping throughout the text the text would be easier for readers to follow. I propose to name groups like Fuc_0, Fuc_0.5, Fuc_1, Fuc_2, corresponding to control, Fuc 0.5%, Fuc 1% and Fuc 2%.
5. Use headings in the discussion section.
6. Line 247 and line 248 show different descriptions under the same number. Please, correct.
Author Response
Your study is showing importance of the dietary modulation of gut microbiota for development of gastrointestinal tract.
Major comments:
- The experimental flow must be presented in form of schematic illustration, showing total amount of animals, division into the groups, introducing the diets, material collection (e.g. for which method etc). Since you use 12 tanks, you have triplicated the groups of larvae for each treatment. Nevertheless, for the results and statistic you are using only 3 animals/group, leading readers to think that you analysed only 1 animal per tank/per group. This number is insufficient for the statistics, results interpretation. If you used 10 animal per groups from 3 different tanks, you should indicate the total amount of animals, e.g. n=30.
Thanks for reviewer’s proposal, and we highly agree that using schematic illustration of the experimental flow would be easier for our readers to follow. We add experimental flow graph (Figure 1) in revised manuscript, page 3 line 109-111. In aquatic animal research, ‘n’ usually represents the number of repeating groups or tanks, not represents the number of animals. For example, we randomly measured final length and weight of 1000 fish larvae per tank, then calculated the average value to compare with average value of other tanks. Previous studies in juvenile fish [1], shrimp [2], and shellfish [3] etc. also use this statistical method strategy to analysis data. Based on helpful advice of reviewer, we will highlight that ‘n’ containing the number of animals in the experimental flow graph.
- Trypsin analysis has been performed in pancreatic tissue and in small intestinal homogenates. Trypsin is produced in form of pro-enzyme (trypsinogen) in the exocrine pancreas, while in small intestine is already activated to its active form. Please, specify what form of trypsin have you analysed.
Thanks for reviewer’s excellent question. Trypsinogen is generally regarded as the inactive precursor of trypsin, and activation of trypsinogen occurs through hydrolyzing the peptide bond. In mammals, the physiological activator of trypsinogen being enteropeptidase that is located on the brush border membrane of enterocytes in the duodenum. However, previous study indicated that trypsinogen possesses proteolytic activity to activate by itself or trypsin in pancreatic tissue [4]. In our manuscript, the principle of trypsin measurement is that trypsin can hydrolyze ester bond of ethyl arginine (substrate), which can be measured under 253 nm ultraviolet light. It seems can’t well distinguish different types of trypsin. However, trypsin activity may hardly response to fucoidan treatment under our experiment condition. Therefore, results of trypsin analysis don't affect the final conclusion that we indicated. Based on helpful comments from reviewer, we will carefully consider different form or tissue of trypsin in our future research.
- I highly recommend to divide Figure 1 into Fig 1A and Fig 1B, showing proliferation/differentiation-related genes and intestinal barrier-related genes, respectively.
Thanks for reviewer’s advice, and we highly endorsed. In revised manuscript, we have divided ‘Figure 1’ into two different parts, page 6 line 221 (Figure 1a proliferation/ differentiation-related genes; Figure 1b intestinal barrier-related genes).
- Quality of Figure 2 and 3 must be improved.
Thank reviewer for helping us to notice some image distortion in our manuscript. We have replaced Figure 2 and 3 with high quality images. We also check all figures and tables in the revised manuscript.
Minor comments:
- Do not use abbreviation in the title. Remove (Fuc), line 2.
Thanks for advice. We have removed ‘Fuc’ in the title, Page 1, line 2.
- Please, use official e-mail addresses as a contact information.
Thanks for advice. I asked all co-auther to offer their college e-mail addresses.
- Do not use word ‘micro’ diets, it would be ok to say ‘diets’ instead.
Thanks for advice. We have directly used ‘diets’ in revised manuscript.
- You use double name for your groups, e.g. Fuc1, Fuc2, Fuc3 or Fuc 0.5%, Fuc 1.00% and Fuc 2.00%). May be introducing only one name at the experimental design and keeping throughout the text the text would be easier for readers to follow. I propose to name groups like Fuc_0, Fuc_0.5, Fuc_1, Fuc_2, corresponding to control, Fuc 0.5%, Fuc 1% and Fuc 2%.
This is particularly good advice, and we accept. Readers may be confused by double name of our experimental groups (fucoidan dosage or group name?). We already have renamed our experimental groups (Fuc0, Fuc0.5, Fuc1, Fuc2, corresponding to Fuc 0%, Fuc 0.5%, Fuc 1% and Fuc 2%).
- Use headings in the discussion section.
Thanks for reviewer’s excellent advice. Readers can easily recognize which parts of results we discussed in the article if we use the subtitle. In resubmitted manuscript, we have used headings to separate our discussion section to three different parts (growth, digestive tract, and gut microbiota). And name of headings: ‘4.1 Dietary Fuc boosted healthy growth of fish larvae’; ‘4.2 Dietary Fuc promoted maturation of digestive tract’; ‘4.3 Dietary Fuc improved gut microbiota of fish larvae’.
- Line 247 and line 248 show different descriptions under the same number. Please, correct.
Thanks for advice. We have corrected descriptions in Table 4 (page 8 line 258-259).
Reference:
- Poolsawat, L.; Li, X.; He, M.; Ji, D.; Leng, X. Clostridium butyricum as probiotic for promoting growth performance, feed utilization, gut health and microbiota community of tilapia (Oreochromis niloticus× O. aureus). Aquacult. Nutr. 2020, 26, 657-670, doi:10.1111/anu.13025.
- Duan, Y.; Zhang, Y.; Dong, H.; Wang, Y.; Zheng, X.; Zhang, J. Effect of dietary Clostridium butyricum on growth, intestine health status and resistance to ammonia stress in Pacific white shrimp Litopenaeus vannamei. Fish Shellfish Immunol. 2017, 65, 25-33, doi:10.1016/j.fsi.2017.03.048.
- CHO, S.H.; PARK, J.; KIM, C.; YOO, J.-H. Effect of casein substitution with fishmeal, soybean meal and crustacean meal in the diet of the abalone Haliotis discus hannai Ino. Aquacult. Nutr. 2008, 14, 61-66, doi:10.1111/j.1365-2095.2007.00505.x.
- Kassell, B.; Kay, J. Zymogens of Proteolytic Enzymes. Science 1973, 180, 1022-1027, doi:10.1126/science.180.4090.1022.

Reviewer 2 Report
The manuscript of Yin et al. investigates the effect of fucoidan supplementation in the diet on body growth, gastrointestinal tract development, and gut microbiota colonization of large yellow croaker`s larvae.
The manuscript is well-written and provides novel insight into how specific nutritional interventions can influence the developmental trajectories of the host and the microbiota in fish. The experiments are in general well-conducted, with sound experimental designs and appropriate methodology.
I have minor comments for the authors (see below).
Comments:
-16s rRNA analysis of the gut microbiota. I suggest better describing the DNA extraction methodology and the 16s rRNA analysis. In particular, the Authors should:
-add the amount of intestinal material used for the DNA extraction
-state if they used MOCK microbial community standards in order to identify taxa that may be over- or under-represented in faecal samples
-specify if any test for kit contamination by using a negative molecular grade water sample
-upload the raw sequence data to a public database.
-Intestinal histology (par. 2.4.1, Methods section): details should be provided on how the intestinal structures were measured. For instance, how the villus height was determined? And the depth of the crypts? These details can be included in Supplementary Figure 2. Also, the magnification used in the imaging should be provided in the manuscript.
-in the Fucoidan treatment groups the intestine looks bigger compared to the control group. This dietary intervention is effective in obtaining bigger fish, however, there is no indication that bigger fish are also healthier. Examining the effect of this prebiotic administration in early life on adult fish could clarify this point.

Author Response
The manuscript of Yin et al. investigates the effect of fucoidan supplementation in the diet on body growth, gastrointestinal tract development, and gut microbiota colonization of large yellow croaker’s larvae.
The manuscript is well-written and provides novel insight into how specific nutritional interventions can influence the developmental trajectories of the host and the microbiota in fish. The experiments are in general well-conducted, with sound experimental designs and appropriate methodology.
-16s rRNA analysis of the gut microbiota. I suggest better describing the DNA extraction methodology and the 16s rRNA analysis. In particular, the Authors should:
-add the amount of intestinal material used for the DNA extraction
-state if they used MOCK microbial community standards in order to identify taxa that may be over- or under-represented in faecal samples
-specify if any test for kit contamination by using a negative molecular grade water sample
This is particularly good advice, and we accept. We have described DNA extraction methodology and the 16s rRNA analysis in detail in revised manuscript, page 4 line 145-152. We didn’t use MOCK microbial community standards or test for kit contamination. However, we believe our 16S rRNA sequencing data are solid and stable, based on two parts. Firstly, dissection of larval gut strictly under sterile and unpolluted environment. We disinfected whole fish body twice by using 75% ethyl alcohol before fish was sacrificed; We dissected intestinal tissue under super-clean worktable and used sterile reagent and centrifuge tube. Secondly, our 16S rRNA sequencing data are not only closely associated with previous study in large yellow croaker [1], but also associated with previous study in marine fish larvae [2]. Based on reviewer’s excellent proposal, we will use MOCK microbial community standards test and contamination test in our future study.
-upload the raw sequence data to a public database.
Thanks for reviewer’s excellent advice. We welcome all readers download our sequencing data in public database, so we had already uploaded clean data in the NCBI Sequence Read Archive (SRA) database (project number: PRJNA735925; uploading date: 8-Jun-2021). Based on helpful comments from reviewer, we also have mentioned sequencing data project number in revised manuscript, page 4 line 161-162.
-Intestinal histology (par. 2.4.1, Methods section): details should be provided on how the intestinal structures were measured. For instance, how the villus height was determined? And the depth of the crypts? These details can be included in Supplementary Figure 2. Also, the magnification used in the imaging should be provided in the manuscript.
Thanks for reviewer’s proposal, and we endorsed. In revised manuscript, we have described the methods that how intestine morphology were measured both in Methods section (page 3, line114-121) and Supplementary Figure 2 (page15 line 447-451). We also have provided the magnification of microscope in revised manuscript, page15 line 448-451.
-in the Fucoidan treatment groups the intestine looks bigger compared to the control group. This dietary intervention is effective in obtaining bigger fish, however, there is no indication that bigger fish are also healthier. Examining the effect of this prebiotic administration in early life on adult fish could clarify this point.
Thanks for reviewer’s excellent question. Analyzing results, we indicated dietary fucoidan improve growth, digestive tract maturation, antioxidant ability, and gut micro-ecology in fish larvae. It seems that fish be healthier to response to dietary fucoidan, especially dietary fucoidan stimulate digestive tract maturation and enhance absorption of nutrients, which resulting in better growth performance of fish larvae. Reviewer pointed out promising research direction that whether early-life intervention by using fucoidan affect health of fish in the adult life. Recently, we also had studied similar research on nutritional programming of large yellow croaker larvae (the effect of early-life nutrients intervention on the adult life) [3]. We believe we will push our research into this field next based on great comments from reviewer.
Reference:
- Wei, N.; Wang, C.; Xiao, S.; Huang, W.; Lin, M.; Yan, Q.; Ma, Y. Intestinal Microbiota in Large Yellow Croaker, Larimichthys crocea, at Different Ages. J. World. Aquac. Soc. 2018, 49, 256-267, doi:10.1111/jwas.12463.
- Borges, N.; Keller-Costa, T.; Sanches-Fernandes, G.M.M.; Louvado, A.; Gomes, N.C.M.; Costa, R. Bacteriome Structure, Function, and Probiotics in Fish Larviculture: The Good, the Bad, and the Gaps. Annu. Rev. Anim. Biosci. 2021, 9, 423-452, doi:10.1146/annurev-animal-062920-113114.
- Liu, Y.; Yao, C.; Cui, K.; Hao, T.; Yin, Z.; Xu, W.; Huang, W.; Mai, K.; Ai, Q. Nutritional programming of large yellow croaker (Larimichthys crocea) larvae by dietary vegetable oil: effects on growth performance, lipid metabolism and antioxidant capacity. Br. J. Nutr. 2022, 1-14, doi:10.1017/s0007114522001726.
